# The Impact of a Child-Friendly Design on Children's Activities in Urban Community Pocket Parks

Le Zhang [1,2], Xiaoxiao Xu [1,2] and Yanlong Guo [3,*]

1   School of Arts, Anhui Jianzhu University, Hefei 230601, China; zhangle@ahjzu.edu.cn (L.Z.); xuxiaoxiao620@stu.ahjzu.edu.cn (X.X.)
2   Anhui Provincial Institute of Culture and Tourism Innovation and Development, Hefei 203106, China
3   Social Innovation Design Research Centre, Anhui University, Hefei 203106, China
*   Correspondence: 20106@ahu.edu.cn; Tel.: +86-152-5655-6306

**Abstract:** Urbanization is a global trend that is expected to continue, and by 2025, it is estimated that almost 60% of the world's children will live in urban areas. Urban community pocket parks provide a solution to the need for parks in high-density urban communities due to their flexible location, small size, and patchy distribution. This paper aims to examine and optimize the construction of urban community pocket parks from a child's perspective to encourage children's participation in these parks. The first step was to conduct a literature review to identify key evaluation indicators for assessing the child-friendliness of pocket parks. Then, the AHP-entropy TOPSIS approach was used to establish an indicator system to effectively evaluate the child-friendliness of pocket parks in urban communities. The system included physical space, cognitive ability, emotional development, environmental perception, and social interaction. Finally, suggestions for optimization were made based on the weighting of influencing factors. The results show that freedom of movement (6.2%) significantly affects the child-friendliness ratings of community pocket parks. Additionally, Hefei residents are not sufficiently influenced by the diversity of play (2.29%) and play facility planning (2.58%) in pocket parks. Therefore, consideration should be given to focusing on the degree of nature adaptation in park construction and renewal projects, as well as understanding children's perception of nature.

**Keywords:** child-friendly environment; pocket parks; evaluation framework; AHP; entropy weights; TOPSIS

## 1. Introduction

In June 2019, the China Community Development Association launched the "Code for Building Child-Friendly Communities in China". According to this code, a neighborhood is defined as a social community of people residing within a certain geographical area [1]. As globalization continues to increase and cities undergo modernization, urban population growth has surged significantly. The United Nations projects that by 2050, 68% of the global population will be concentrated in urban centers. With this pace of urbanization, it is expected that, by 2025, urban areas will be home to around 60% of the world's children. However, there is a significant tension between children and cities due to the growing size of cities and the increasing demands for quality of life. Therefore, promoting child-friendly cities is essential for providing good quality of life and accessibility for all children in urban areas. This is especially important in public spaces such as pocket parks, where children are most active. Child-friendly design in such spaces can create a safe, enjoyable, and stimulating environment that promotes sustainable physical activity for children [2].

A pocket park is defined as a park that is smaller than most urban parks [3], and the definition of pocket park size varies from region to region, from less than half an acre to less than 5000 square meters [4]. American scholar Robert Zane designed the world's first pocket park in 1963, marking the birth of the micropark out of which the prototype was

a small park scattered in high-density urban centers and arranged in patches for the use of locals. As socio-economic development and population mobility intensified, pocket parks gradually became a new urban form. The earliest Chinese definition of a pocket park is by Zhang Wenying (2007): a community pocket park is a small, open space of greenery that is scattered in patches or hidden within the urban infrastructure [5]. According to scholars' definitions of pocket parks, their characteristics can be summarized as: (1) smaller in size than the average urban park, and (2) scattered in urban communities. In earlier times, scholars primarily focused on the advantages, purposes, and utilities of green spaces or urban parks [6–8]. Few studies were conducted on them until 2010, after which they were gradually enriched to be focused on more generally, with the emergence of a focus on other types of infrastructure such as rooftop-type green spaces [9]. As people's lifestyles evolve, research has demonstrated that residents in residential areas tend to favor smaller parks over larger urban parks [10]; having green spaces in places within walking distance of home or work can have a positive impact on resident satisfaction and frequency of use [11]. Community pocket parks act as a small public space within the community, taking a variety of forms on the side of streets and corners, at community entrances and exits, often as a vibrant medium for everyday life. Community pocket parks have evolved from being a mere "niche" concept to becoming a crucial aspect of daily recreation and a significant tool for promoting innovation in urban social governance.

However, in some urban community pocket parks, there are relatively few places and facilities for children's activities, and children's access to parks is often inequitable in terms of distance and quality, and lacks a child-friendly design [12]. This has implications for children's physical and mental health and development, such as a lack of suitable places for exercise and play, a tendency for children to become addicted to electronic devices, and a lack of social interaction. Children are an easily overlooked and underserved population in park planning, design, and management decision-making [13,14]. Children need to be involved in urban planning and design, and as important users of community parks, they need to be involved in their design [15]. Their design indicators enhance the child-friendliness of the whole park while meeting the use needs of other age groups in accordance with the children's scale. A park is a park for all, and when we make a park for all, we are also making a park for children [16]. After reviewing 25 studies across 11 sources, Chinese scholar Xue Meng identified several factors that influence child-friendly pocket parks. These factors can be broadly classified as follows: the environment and its accessibility, the perceived level of safety and opportunities for physical activity and recreation, and the visual attractiveness and comfort of the surrounding area [17].

Surroundings and accessibility: child-friendly elements in urban spaces should prioritize surroundings and accessibility, which include factors such as ease of access, perceived spatial depth, visual range, height, and a diverse range of colors [18]. Accessibility to urban parks refers to the accessibility and ease of access to different parks for residents returning home, children after school or elderly people chatting, and aims to measure the 'opportunity potential' outside the park. Accessibility has been measured using urban parks as units of public service provision, with buffer zone analysis, network analysis, urban road network topology analysis and other methods [19]. The third method usually uses spatial syntactic theories and models, with the help of graph theory, and provides a mathematical and quantitative description of the spatial structure patterns and their organization laws, and measures the park's accessibility by calculating the integration and selection degrees [20].

Activity and perceived safety: safety is undoubtedly an important issue for children [21]. But in this context, excessive safety is a social pathology that can turn into a new wave of child spoiling. There are a number of factors that can create problems for children's safety in parks, such as the way in which play is generated by dangerous activity spaces in parks. Safety issues in play spaces are often attributed to the excessive use of hard materials. To address this, loose materials such as tires, wooden planks, and plastic crates can be incorporated into the design of play spaces to enhance safety during play

while still allowing children to explore and be creative [22,23]. In addition, the choice of facilities with too much plastic in the space can also lead to a risk of microplastics being inhaled by children [24], the above being factors caused by the environment of the activity space. Smaller pocket parks also have the potential to increase the opportunity for conflict due to the concentration of play facilities, such as strangers from different communities who may have a negative impact on children due to cultural differences [25].

Children's sports and play opportunities: the factors that impact children's opportunities for sports and play can be summarized as the diversity of parks available [26,27]. Park facilities are an important element in building child-friendly cities [28]. Furthermore, the availability of play opportunities and the quality of play facilities are significant concerns. In many cases, parks are overcrowded due to inadequate maintenance, lower safety standards, and a higher prevalence of physical environmental hazards, which can negatively impact the quality of play experiences [29–32]. The potential for psychological recovery can be better provided by well-designed miniparks [33]. A good-quality pocket park is one that allows residents to achieve mental and physical well-being during the park experience while completing a number of recreational activities [34]. When evaluating children's access to outdoor play opportunities, it is crucial to prioritize access to parks that offer safe and high-quality play facilities, rather than simply striving for access to every park indiscriminately. In other words, it is not enough to just have access to any park—the quality and safety of the play facilities within the park are critical factors to consider when evaluating its suitability for children's play.

Aesthetics and comfort of the environment: research on the aesthetics and comfort of outdoor environments has typically emphasized the role of thermal sensations as an indicator of actual comfort [35]. Nonetheless, evaluating outdoor comfort is inherently complex, as it involves multiple contextual factors within open spaces, the socio-demographic characteristics of individuals, and psychological factors [36]. The factors that influence the comfort of a park environment may manifest in different ways across various sections of the park. This means that each area of the park may have unique considerations that impact its overall comfort level, and must be evaluated accordingly [37]. Plants are considered to be one of the most important components influencing the viewer's perception of the environment [38]. Natural plants in parks that are otherwise toxic and harmful need to be transformed, but not over transformed into a plastic version of nature, filled with overly fake and overly decorative plants. The original spiritual, vibrant quality of nature almost disappears, and such parks do not benefit children.

Chinese scholars are gradually enriching their research on "child-friendly" parks, but in general, it seems that the research is mostly confined to the design of children's playgrounds, and there are fewer studies on the systematic evaluation of child-friendly parks, and the design specifications and methods of child-friendly parks are not yet systematic. As the construction of 'child-friendly' cities in China continues to advance, parks and green spaces are an integral part of cities, and pocket parks are an important place for children to spend time outdoors, so exploring how to design 'child-friendly' pocket parks in a systematic and standardized way is a pending issue. This article presents the findings of this study through fieldwork and research on the literature. This article sums up the evaluation indexes of the child-friendliness of pocket parks through fieldwork and research on the literature, and constructs an index system to assess the child-friendliness of pocket parks in urban communities, including physical space, cognitive ability, emotional development, environmental perception, and social interaction. According to the weighting of the influencing factors, the corresponding optimization strategies are proposed. This study has certain implications for the renewal and optimization of child-friendly community pocket parks.

## 2. Study Site and Method

### 2.1. Study Site

The research conducted in this article centers around the city of Hefei in Anhui Province, which is projected to have a population of 9.465 million by the end of 2022, with an urbanization rate of 84.64%. Hefei is actively pursuing the development of "pocket parks" and is working to increase the amount of green space available to local residents. As of now, Hefei has already established over 175 pocket parks, and plans to add an additional 150 more within the next three years, starting from 2021. The study focuses on areas of high pedestrian traffic within Hefei's urban center, as shown in Figure 1. The green areas in the figure represent existing parks in Hefei. The chosen urban area in the center of Figure 1 has a high density of green spaces, with a green space system characterized by point and linear green spaces, including small parks and green belts located along the river system and roads. We can limit the adverse effects of accessibility issues on urban parks to some degree by focusing on this particular area, while also taking into account the compact size of the green space and ensuring that it meets the criteria for a pocket park.

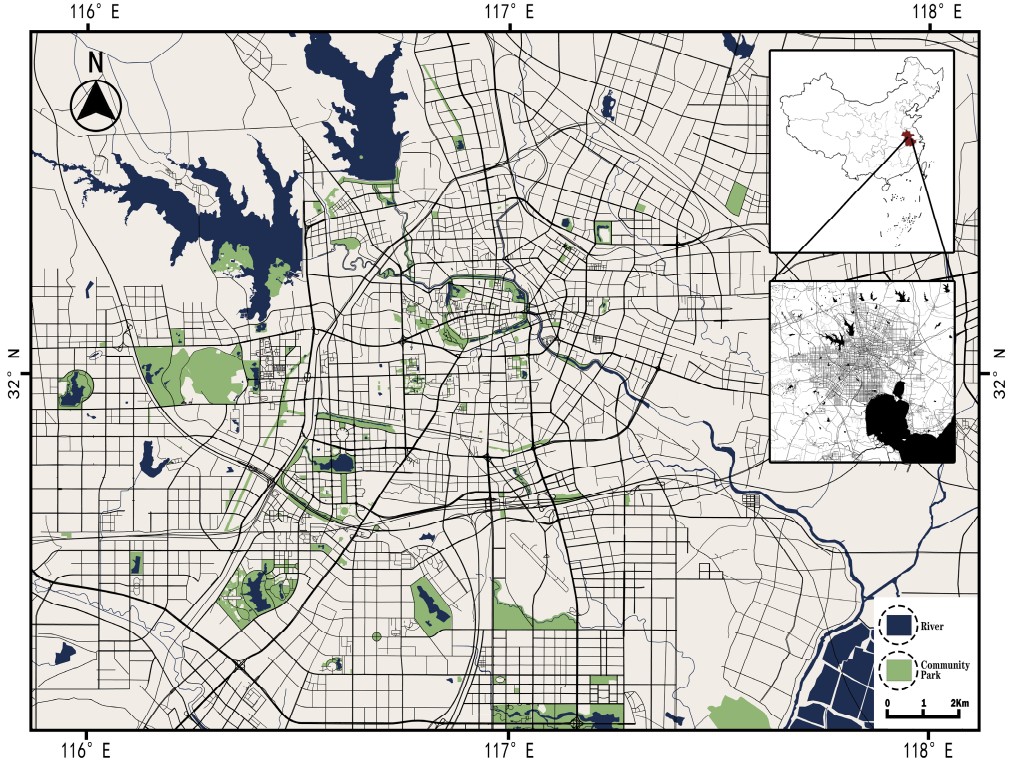

**Figure 1.** Locations of community pocket parks in Hefei, Anhui, China.

Field visits to urban community pocket parks in Hefei as well as child-friendly analyses were conducted to select those with a high degree of relevance for analysis.

Green Axis Park, also known as Hefei's version of Central Park, was completed in 2016. It is located in the core of Hefei's Shushan District, south of the main axis of the Government Affairs Centre. The park is the southern center of the new district's north-south central green space and is planned to occupy a total area of 18.84 hectares. However, this field study selected an area of 9700 square meters that was renovated and upgraded in 2019 for investigation.

The renovated area is divided into four main areas: the forest oxygen bar, the football tribe, the extreme world, and the children's playground. The renovation included the construction of sports facilities such as a five-a-side football pitch, basketball court, parkour sports area, skateboard park, and children's development area. Additionally, functional facilities such as square open space, garden path repair, and signage improvement

were also improved. The park's greening was also enhanced. The renovation focused on improving the safety protection and warning information of the skateboard park, children's development area, and other sports facilities to ensure the public's safety during recreational sports.

Green Axis Park incorporates the concept of humanism and evenly distributes facilities of different functional subjects in the park. This ensures the sports and leisure functions of the park while allowing the public to feel the humanistic care within the park. The transformation of the park meets the diversified needs of the public and creates an urban ecological park with sports, leisure, culture, and humanism based on the concept of science, ecology, and environmental protection. The park provides a wonderful place for the public to enjoy (Figure 2).

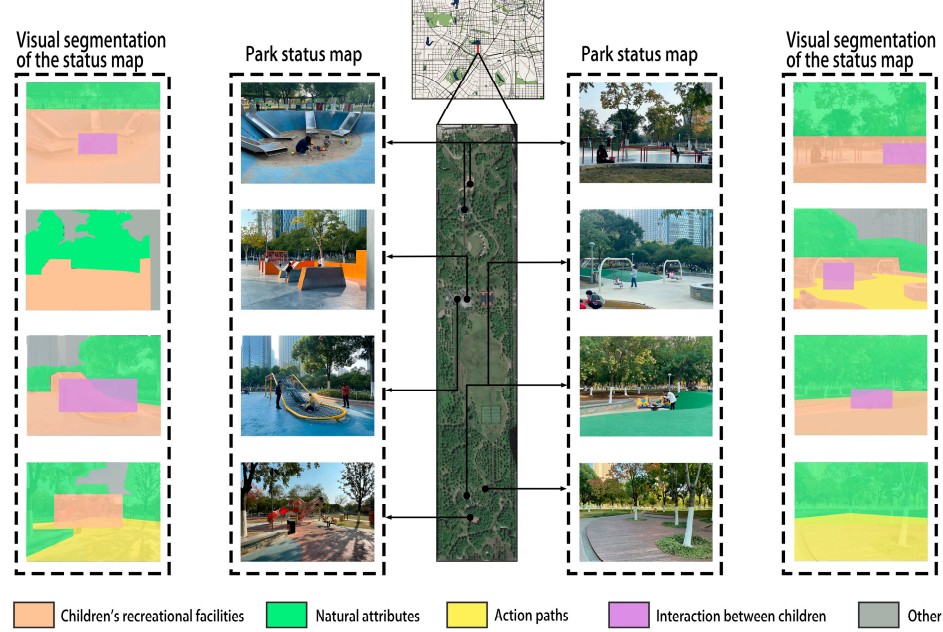

**Figure 2.** Green Axis Park child-friendly analysis.

Shitai Road Park is located at the southeast corner of the intersection of Swan Lake East Road and Shitai Road in Shushan District, Hefei. The park has a total area of 68,900 square meters and is divided into various areas such as the rhythm leisure area, diversity lawn area, fun activity area, and dream highland area to meet the leisure needs of different groups.

One of the park's main features is its "intelligent" design, with a large intelligent guide screen that allows residents to learn about the park's attractions with just a click. The park has also installed the "Frog in the Lotus" intelligent robot, which interacts with people and frogs through an AI intelligent voice dialogue system. Additionally, interactive musical swings and jumping springs have been installed to increase interactivity and fun. The park adopts a three-dimensional composite transportation system through setting up intelligent fitness tracks, recreational trails, and forest walkways to meet functional needs. Aerial facilities such as aerial stacks and viewing bridges have also been set up to form an up-and-down composite transportation system, allowing visitors to enjoy the park's beauty in a 360° manner.

With its intelligent and technological features, unique landscape design, and greenery planting, Shitai Road Park has become an important place for leisure, entertainment, and cultural exchange in Shushan District. It provides a wonderful place for the public to enjoy (Figure 3).

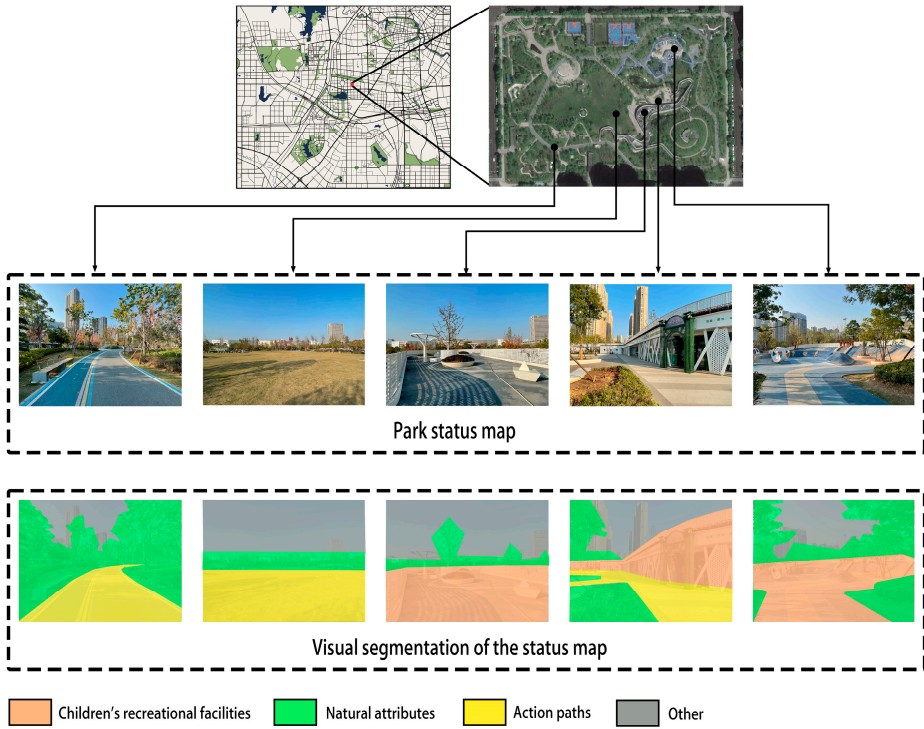

Children's recreational facilities   Natural attributes   Action paths   Other

**Figure 3.** Child-friendly analysis of Shidai Road Park.

*2.2. Construction of Evaluation Indicators*

The assessment of child-friendliness in city neighborhood pocket parks needs to be completed by way of sure indicators, and Mohit Kumar Agarwal used drawing as a methodological device to understand perceive kids' expectations and to ascertain their preferences for a perfect neighborhood park. Five predominant warning signs emphasizing physical, perceptual, cognitive, emotional, and social parameters were derived to enhance child-friendly environments in neighborhood public areas [39]. The contrast indications are of some referential significance, and a contrast indicator device was developed on this basis, taking into ac-count the Chinese context (Table 1).

Important indicators that should be considered for the child-friendliness of pocket parks include:

(1) Physical space: The ecological mannequin states that kids' fitness is influenced by the traits of their bodily surroundings [40]. In particular, challenging and exciting play gear can generate distinct ideas for children, as well as open spaces with treasured fitness features [41]. During fieldwork, it was discovered that the spatial scale of the park correlated with an increase in older young people's choice to visit, and their preference for fitness services and facility accessibility modified with age, as validated through research with the aid of Elliott P. Flowers [42]. The normative dimension of physical parameters in the theoretical framework for child-friendly environments encompasses factors such as the environmental quality and aesthetic appeal of the city, among other dimensions [43]. The area of green space per capita and the distribution of pocket parks in the community is also used as one of the physical parameters to evaluate child-friendliness due to their flexible locations, small size, and patchy distribution.

(2) Cognitive skills: Children boost cognitive competencies via exploration and experience of the social, bodily, and natural environment [44]. They are tremendously attracted to exploratory play and activity areas will supply them the chance to locate and meet new pals [45]. Research has proven that the play points and range of current services in the park are essential for growing visitation, i.e., the range of things to do in the pocket park [41]. On this basis, pocket parks are closest to the daily communi-

cation, interaction, and level of connection of citizens and can provide a good place for environmental education to take place, with a significant impact on the duration of children's physical activity [46]. There are differences in educational needs for environmental education in terms of site planning and practical activities in different population information conditions [47].

(3) Emotional development: Children have unique perceptions of their surroundings in contrast to adults. Different environmental elements can elicit excessive emotional responses from young people [48]. Children have fewer opportunities to journey their environment alone, which contributes to their loss of focus of environmental safety, safe-ty being one of the most vital facets in designing child-friendly environments, and lack of protection and safety leads to a large reduction in the amount of outdoor things kids' can do [48]. Outdoor play fosters social-emotional guide structures [49]. Children's emotional development can be influenced in distinct ways by various types of play. Wide and open spaces provide the opportunity to run freely and release their inner energy, which can significantly affect children's social anxiety [50]. Emotional affinity with nature and time spent in nature during childhood can effectively nurture a sense of belonging to a place [51].

(4) Environmental perception: Exploring child-oriented public areas in different neighborhoods from an environmental psychology point of view to improve kids' environmental perception can help to enhance the home and surroundings in which youngsters develop [52]. Children perceive play areas and open spaces as opportunities for play and interaction with nature. That is why playgrounds that are carefully designed and planned can provide children with a variety of options to satisfy their diverse needs and desires [52]. The natural organic attributes of the park have a strong effect on the way young people learn, specially via play [53]. During the fieldwork, it was discovered that young people cited bushes and grassy open areas as their preferred section of parks, and that interacting with plant life in the park and experiencing specific physical sensations can be an important quality in improving their understanding of natural surroundings in the context of nature conservation and protection for self-generation [54].

(5) Social interaction: In pocket parks, play can promote the development of children's personalities. One indicator of this is social competence, which is a measure of children's ability to assemble and includes the likelihood of children engaging in social interaction [55]. An observational study of Australian children showed that 85% of responses indicated that children played or interacted with others (e.g., peers, parents, or siblings) [56]. Research has shown that the absence of friends can impede children's engagement in physical activity, emphasizing the importance of socializing and interacting with children of different ages [57]. Through field surveys, the size of public recreational spaces was found to be a critical factor in establishing a child-friendly environment. The size of these spaces plays a pivotal role in determining the likelihood of social interactions with peers of varying ages within the same area.

**Table 1.** Evaluation criteria system for child-friendly design in community pocket parks.

| Target Level | Guideline Level | Programmer Level |
|---|---|---|
| A: Evaluation of the child-friendliness of urban community pocket parks | B1: Physical space [40] | C1: Accessibility of facilities [42] |
| | | C2: Openness in the park [58] |
| | | C3: Walkability in the park [58] |
| | | C4: Environmental quality [43] |
| | | C5: Park surface green space area |
| | | C6: Green space per capita |
| | | C7: Number of parks |

**Table 1.** *Cont.*

| Target Level | Guideline Level | Programmer Level |
|---|---|---|
| | B2: Cognitive skills [47] | C8: Diversity of play [41] |
| | | C9: Challenging to explore [45] |
| | | C10: Significance of learning [47] |
| | B3: Emotional development [48] | C11: Health and safety [48] |
| | | C12: Freedom of movement [49] |
| | | C13: Nature affinity [51] |
| | | C14: Length of time spent in nature [50] |
| | | C15: Urbanization rate |
| | B4: Environmental perception [52] | C16: Design aesthetics [52] |
| | | C17: Play facility planning [52] |
| | | C18: Diversity of natural attributes [53] |
| | | C19: Air quality (proportion of days with air quality at and better than Class 2) |
| | B5: Social interaction [55] | C20: Shareability [56] |
| | | C21: Public recreational space |
| | | C22: Participatory [57] |
| | | C23: Age structure of children (% aged 0–14) |

*2.3. Formatting of Mathematical Components*

There are two main categories for determining impact factors: subjective and objective methods. Subjective methods for evaluating the child-friendliness of urban communities typically involve the evaluator's subjective focus on relevant factors and the actual situation, often with input from experts in the field. Subjective methods such as hierarchical analysis and statistical analysis are commonly used, while objective methods rely on impact factors to assign weight, such as entropy weighting.

Weighting methods that are subjective require the evaluator to subjectively determine the relative importance of each factor, which can be arbitrary and lack objectivity. In contrast, objective weighting methods have a strong mathematical basis and avoid subjective influence, but may not always align with reality or the evaluator's intentions.

To mitigate the influence of experts' empirical knowledge, the AHP method selects scores from multiple comparisons at various levels and tests the consistency of judgment matrices. The AHP weights are obtained by combining the scores using the arithmetic average of weights. However, objective assignment methods can sometimes place excessive emphasis on the internal variation between the data of each evaluation indicator.

To address these concerns, this study integrates both subjective and objective assignment methods. Weights are determined by separately applying the AHP method and entropy weighting method, and then combining the results. This approach seeks to achieve a more balanced and objective assessment of the child-friendliness of urban communities. This yields comprehensive weights that are more credible [59,60]. See Figure 4 for specific data processing methods.

2.3.1. AHP Method to Calculate Indicator Weights

Saaty developed the AHP method in the 1970s as a means of assigning weights to alternatives in decision-making [61]. By combining the strengths of the analytic hierarchy process and the fuzzy evaluation method, this approach allows decision makers to assign specific weights to each potential solution [62]. This approach allows decision makers to

choose the attributes that align best with their business development needs, empowering them to make more informed and effective decisions.

| Step 1 | Step 2 | Step 3 | Step 4 |
|---|---|---|---|
| Questionnaire data processing | Data normalisation process | AHP calculation weights | Entropy weight calculation weights |
| Questionnaire data reliability tests were conducted to determine the reliability of the questionnaire data. | Normalization of question-naire data with yearbook data. | The normalized data were used for AHP weight calculation as well as to test the consistency of the judgment matrix. | Normalized data for entropy weighting |

| Step 5 | Step 6 | Step 7 | Step 8 |
|---|---|---|---|
| Combined weighting calculation | TOPSIS method analysis | TOPSIS method evaluation | Comprehensive evaluation of results |
| AHP weights and entropy weights for combined weight calculation | Calculation of optimal and inferior solution evaluation objects | Compare the relative proximity of each evaluation object to the positive understanding | Comprehensive evaluation of the combined weights and TOPSIS calculation results |

**Figure 4.** Data processing steps.

This study introduces a hierarchical AHP method that evaluates and ranks expert opinions based on sublevels to address decision-making problems. It recommends pairwise comparisons of linguistic variables on a nine point scale. In every set of comparisons, the AHP algorithm calculates the geometric mean, arithmetic mean, and integral between each comparison matrix. As a result, the AHP is a dependable and user-friendly method for assigning relative weights to sustainability assessment tool indicators [63].

(1) Assume that the decision problem can be built to include $n$ indicators, evaluation indicators can be sorted into $i$ categories, one of which can be divided into $j$ indicators; the weight assigned to each indicator, denoted by $T_{ij}$, is calculated using either the "1 to 5 scale" method or the automatic adjustment method, and reflects the degree of influence that indicator $j$ has on indicator $i$. To determine the relative weight of lower-level indicators on upper-level indicators, a comparison matrix is constructed for each indicator layer. This allows for a systematic evaluation of the importance of each indicator in relation to the others at its level, ultimately leading to an assessment of the overall significance of each indicator in contributing to the higher-level objective.

$$T_{ij} = \begin{bmatrix} t_{11} & t_{12} & \ldots & t_{1j} \\ t_{21} & t_{22} & \ldots & t_{2j} \\ \vdots & \vdots & \ddots & \vdots \\ t_{j1} & t_{j2} & \ldots & t_{jj} \end{bmatrix} \tag{1}$$

(2) A test for consistency is conducted to evaluate the judgment matrix that has been constructed.

$$CI = \frac{\lambda_{max} - s}{s - 1} \tag{2}$$

$$CR = \frac{CI}{RI} \tag{3}$$

To evaluate the consistency of the judgment matrix, the consistency index (CI) is calculated; the order of the matrix (s), the matrix's eigenvalue ($\lambda_{max}$), the consistency ratio (CR), and the average random consistency index (RI). A consistency test is performed by

comparing the CR to a threshold value of 0.10. If the consistency ratio (CR) is below 0.10, the judgment matrix is considered consistent. If the CR exceeds 0.10, the matrix needs to be adjusted to enhance its consistency.

### 2.3.2. Entropy Weighting Method for Calculating Indicator Weights

Entropy is a metric used to quantify the level of disorder or randomness in a system. An increase in entropy leads to a decrease in the amount of useful information, while a decrease in entropy results in an increase in the amount of useful information. The entropy weighting approach is applied to assign weights to each evaluation index. In this method, a higher weight is assigned to an index that utilizes a greater amount of effective information in its calculation [64]. The entropy weighting method is computed using the following steps:

(1) Build a judgment matrix of $n$ samples and $m$ evaluation factors $R = (\chi_{ij})_{nm} (i = 1, 2, \dots, m; j = 1, 2, \dots, n)$

(2) Normalize the judgment matrix to obtain a matrix of normalized values $y = (y_{ij})_{nm}$

$$y_{ij} = \frac{\chi_{ij} - \chi_{min}}{\chi_{max} - \chi_{min}} \tag{4}$$

In the equation, $y_{ij}$ represents the value of the element in the $i$ row and $j$ column of the matrix $y$, while $\chi_{ij}$ represents the measurement of the $j$ evaluation indicator for the $i$ sample. Additionally, $x_{min}$ refers to the minimum value observed across different samples for the same indicator, while $x_{max}$ denotes the maximum value recorded across different samples for the same indicator. These values are used to normalize the judgment matrix R and obtain the corresponding normalized matrix $y$.

(3) To calculate the entropy and entropy weight of indicator $j$, the following steps can be taken:

$$H_j = -\frac{1}{lnn} \left( \sum_{i=1}^{n} f_{ij} ln f_{ij} \right) \tag{5}$$

$$f_{ij} = \frac{1 + y_{ij}}{\sum_{j=1}^{n} + y_{ij}} \tag{6}$$

$$w_i = \frac{1 - H_j}{m - \sum_{j=1}^{m} H_j} \tag{7}$$

where, in this context, $H_j$ represents the entropy of the indicator, $f_{ij}$ refers to the element found in the $j$ column and $i$ row of the matrix, and $w_j$ denotes the entropy weight of the indicator.

### 2.3.3. Combined Weighting Method

Equation (8) represents the combination of weights calculated using both the AHP and entropy methods, which allow for the calculation of both subjective and objective weights. The combined weighting method involves utilizing the AHP method to determine subjective assignment weights, and the entropy method to determine objective assignment weights. The combined weights are then calculated using the following equation:

$$W_i = \frac{W_{AHP} W_{Entropy}}{\sum_{i=1}^{n} W_{AHP} W_{Entropy}} \tag{8}$$

where: $W_i$ is the combined weight; $W_{AHP}$ is the subjective weight of the factor; and $W_{Entropy}$ weight is the objective weight of the factor.

### 2.3.4. AHP-Entropy TOPSIS Evaluation Model

The TOPSIS (Technique for Order of Preference by Similarity to Ideal Solution) method is an approximate ideal solution ranking method that involves evaluating the closeness

between the object being evaluated and an idealized target in order to determine their relative merits [65]. The TOPSIS method involves defining a set of evaluation criteria or attributes, and then calculating the distance between the evaluated object and the idealized target for each attribute. The idealized target represents the best possible performance for each attribute, while the evaluated object represents the actual performance. The distances are then used to determine the relative merit of each evaluated object, with those closest to the idealized target receiving the highest ranking. The TOPSIS method is widely used in decision-making processes due to its simplicity and ease of implementation [66]. The TOPSIS method has the following main steps.

(1) The original data matrix $X$ is derived from the evaluation objects and evaluation indicators. Let there be $m$ evaluation objects ($X_1$, $X_2$, ..., $X_m$) and $n$ evaluation indicators (1, 2, ..., $n$) for a decision problem.

$$X = \begin{bmatrix} X_{11} & X_{12} & \cdots & X_{1n} \\ X_{21} & X_{22} & \cdots & X_{2n} \\ \vdots & \vdots & \ddots & \vdots \\ X_{m1} & X_{m2} & \cdots & X_{mn} \end{bmatrix} \tag{9}$$

(2) To obtain the normalization matrix $F$, the original indicators are first normalized. Let $x_{ij}$ denote the value of the $j$ indicator for the $i$ scheme in the $X$ matrix, and let $f_{ij}$ denote its normalized value. The normalization process involves transforming the values of each indicator to a common scale, typically between 0 and 1, to ensure that all indicators are given equal weight in the subsequent analysis. The resulting normalization matrix F contains the normalized values of all indicators for each scheme, allowing for a comparative evaluation of their performance.

$$f_{ij} = x_{ij} / \sum_{i=1}^{m} x_{ij} (i = 1, 2, \cdots, m; j = 1, 2, \cdots, n) \tag{10}$$

$$F = \begin{bmatrix} f_{11} & f_{12} & \cdots & f_{1n} \\ f_{21} & f_{22} & \cdots & f_{2n} \\ \vdots & \vdots & \ddots & \vdots \\ f_{m1} & f_{m2} & \cdots & f_{mn} \end{bmatrix} \tag{11}$$

(3) To construct a weighted normalization matrix $R$, to assign weights to each indicator, the AHP-entropy weighting method is employed, and the weighted normalization matrix is constructed using the normalization matrix $F$. Let $r_{ij}$ denote the value of the $j$ indicator for the $i$ scheme in the weighted normalization matrix $R$. The weights of each indicator are determined using the AHP-entropy weighting method, which combines subjective and objective weights to provide a more balanced evaluation of the indicators. The resulting weights are then used to construct the weighted normalization matrix $R$, which accounts for both the relative importance of each indicator and the normalized values of each scheme's performance on each indicator.

$$r_{ij} = w_j f_{ij} (i = 1, 2, \cdots, m; j = 1, 2, \cdots, n) \tag{12}$$

(4) Determine the sets of positive and negative ideal solutions, denoted by $R^+$ and $R^-$, respectively. For indicators where larger values represent better performance and smaller values represent higher costs, $R^+$ is constructed using the maximum values of the efficiency indicators and the minimum values of the cost indicators, while $R^-$ is constructed using the minimum values of the efficiency indicators and the maximum values of the cost indicators. Thus, $R^+ = \left( r_1^+, r_2^+, \cdots, r_n^+ \right)$ and $R^- = \left( r_1^-, r_2^-, \cdots, r_n^- \right)$.

(5) Calculate the distance of each solution from the positive and negative ideal solutions, denoted by $D^+$ and $D^-$.

$$\begin{cases} D_i^+ = \sqrt{\sum\limits_{j=1}^{n} (r_{ij} - r_j^+)^2} \\ D_i^- = \sqrt{\sum\limits_{j=1}^{n} (r_{ij} - r_j^-)^2} \end{cases}, (i = 1, 2, \cdots, m; \ j = 1, 2, \cdots, n) \tag{13}$$

(6) To assess the relative efficiency or performance of each solution, the proximity to the ideal solution is calculated. The proximity of each solution is denoted by $C_i$, where a larger value of $D_i^+$ indicates that the solution is closer to the optimal solution and is therefore better, while a smaller value of $D_i^-$ indicates that the solution is further away from the optimal solution and is therefore worse. The proximity of each solution $i$ is calculated by dividing the distance from the negative ideal solution by the sum of the distances from the positive and negative ideal solutions, according to the following formula:

$$C_i = D_i^- / (D_i^+ + D_i^-)(i = 1, 2, \cdots, m; \ j = 1, 2, \cdots, n) \tag{14}$$

*2.4. Formatting of Mathematical Components*

The 23 evaluation indicators for child-friendliness were compiled through a combination of field surveys and literature reviews conducted in Hefei. Subsequently, a child-friendly questionnaire survey was administered in the community pocket parks of Hefei to evaluate the performance of the 23 indicators. This survey provided valuable insights into the child-friendliness of the community pocket parks and informed the subsequent analysis and decision-making processes. The interviewees were required to be children aged 0–14 years or accompanied by their families in lieu of interviewees to quantitatively evaluate the indicators in Table 2. Table 2 presents the results of a questionnaire that used a scale format. Each question in the scale contained a series of statements, with five possible answers that corresponded to degrees of evaluation. Scores ranged from 1 to 5. Each score in the questionnaire survey reflects the respondent's assessment of the statement being evaluated. The respondents were asked to rate each statement based on their perception of its relevance and importance to child-friendliness, with higher scores indicating greater importance and relevance, and lower scores indicating lesser importance and relevance. These scores were then used to evaluate the performance of the community pocket parks across the 23 evaluation indicators, providing a basis for subsequent analysis and decision making. C5: spatial scale, C6: green space per capita, C7: number of pocket parks, C15: urbanization rate, C19: air quality, and C23: age structure of children were calculated from the available literature, statistical bulletins, and internet information due to the limitations of the questionnaire; therefore, the data for the study was obtained from the existing literature, statistical bulletins, and internet information. The questionnaire consisted of basic information (gender, age, education, age of children, and community pocket parks) and an evaluation of the child-friendliness of the community pocket parks. The questionnaire was administered on site by subject members with expertise in interviewing people who met the requirements, and the distribution point was located in a community pocket park space in Hefei.

**Table 2.** Quantification of child-friendly evaluation indicator system in urban communities.

| Programmer Level | Quantification of Indicators |
|---|---|
| C1: Accessibility of facilities | Extremely inconvenient = 1; Inconvenient = 2; Average = 3; Convenient = 4; Extremely convenient = 5 |
| C2: Openness in the park | Extremely inaccessible = 1; Inaccessible = 2; Fair = 3; Open = 4; Extremely open = 5 |
| C3: Walkability in the park | Extremely poor = 1; Poor = 2; Fair = 3; Good = 4; Extremely good = 5 |
| C4: Environmental quality | Extremely poor = 1; Poor = 2; Fair = 3; Good = 4; Extremely good = 5 |
| C5: Park surface green space area | Statistics |
| C6: Green space per capita | Statistics |

**Table 2.** *Cont.*

| Programmer Level | Quantification of Indicators |
|---|---|
| C7: Number of parks | Statistics |
| C8: Diversity of play | Very little = 1; Less = 2; Fair = 3; More = 4; A lot = 5 |
| C9: Challenging to explore | No challenge = 1; Somewhat challenging = 2; Challenging = 3; Very challenging = 4; Very challenging = 5 |
| C10: Significance of learning | No meaningful = 1; Somewhat meaningful = 2; Meaningful = 3; Very meaningful = 4; Very meaningful = 5 |
| C11: Health and safety | Very insecure = 1; Insecure = 2; Fair = 3; Secure = 4; Very secure = 5 |
| C12: Freedom of movement | Very unfree = 1; Not free = 2; Average = 3; Free = 4; Very free = 5 |
| C13: Nature affinity | Very unaffectionate = 1; Unaffectionate = 2; Average = 3; Affectionate = 4; Very affectionate = 5 |
| C14: Length of time spent in nature | Very little = 1; Less = 2; Fair = 3; More = 4; A lot = 5 |
| C15: Urbanization rate | Statistics |
| C16: Design aesthetics | Extremely poor = 1; Poor = 2; Fair = 3; Good = 4; Extremely good = 5 |
| C17: Play facility planning | Extremely poor = 1; Poor = 2; Fair = 3; Good = 4; Extremely good = 5 |
| C18: Diversity of natural attributes | Very little = 1; Less = 2; Fair = 3; More = 4; A lot = 5 |
| C19: Air quality (proportion of days with air quality at and better than Class 2) | Statistics |
| C20: Shareability | Extremely poor = 1; Poor = 2; Fair = 3; Good = 4; Extremely good = 5 |
| C21: Public recreational space | Extremely small = 1; Small = 2; Fair = 3; Large = 4; Extremely large = 5 |
| C22: Participatory | Extremely poor = 1; Poor = 2; Fair = 3; Good = 4; Extremely good = 5 |
| C23: Age structure of children (% aged 0–14) | Statistics |

## 3. Results and Analysis

The questionnaire survey yielded 349 valid responses out of 360 distributed, resulting in an effective rate of 96.94% (Table 3). The age distribution of the children surveyed was as follows: 39.54% for 0–4 years, 29.23% for 5–8 years, and 31.23% for 9–14 years. To ensure the data's reliability and validity, factor analysis was conducted as a method of data analysis to evaluate the reasonableness and practical significance of the research items. KMO values, commonality, variance-explained values, and factor loading coefficient values were analyzed to determine the data's validity level. The analysis revealed that different researchers used varying analysis methods for the same question, leading to some variation in data reliability. KMO values were used to evaluate the extraction of information, while commonality values were employed to eliminate any unreasonable research items. Additionally, Cronbach's alpha coefficient was used to assess the internal consistency of the obtained factors, with a coefficient value of 0.70 or higher indicating good data reliability. A KMO value between 0.8 and 1.0 indicated sufficient sampling. The data were analyzed using statistical software SPSSAU, and the results showed a high level of reliability and quality of the research data, with a Cronbach's alpha coefficient of 0.947. The sample data test statistic KMO was 0.958, with a significance level of $p < 0.05$, indicating that the data passed the validity test and met the necessary criteria for factor analysis (Table 4).

**Table 3.** Cronbach reliability analysis.

| Number of Items | Number of Samples | Cronbach's Coefficient |
|---|---|---|
| 17 | 349 | 0.947 |

**Table 4.** KMO and Bartlett tests.

| KMO | | 0.958 |
|---|---|---|
| **Bartlett** | Approximate cardinality | 4558.56 |
| | df | 136 |
| | *p* value | 0.000 |

### 3.1. Data Normalization

In Table 3, 17 indicators can be obtained from the questionnaire data, while the remaining 6 indicators need to be calculated from the statistical data of the Anhui Provincial Yearbook. However, as the units of the yearbook data and the questionnaire data are different, the relative weights cannot be calculated, so data normalization is needed. Normalization is one of the methods of data standardization, converting dimensioned data to dimensionless data to obtain new data ranging between [0, 1]. The questionnaire data were normalized by importing the questionnaire data into SPSSAU with C5: spatial scale, C6: green space per capita, C7: number of pocket parks, C15: urbanization rate, C19: air quality, and C23: child age structure of the Anhui Province Yearbook 2012–2021 data through Equation (4).

### 3.2. AHP-Entropy TOPSIS to Determine Weights

#### 3.2.1. AHP Determination of Weights

The data collected from the questionnaire survey were used to calculate the AHP weights for the 23 evaluation indicators using Equation (1) (Table 5). A 23-order judgment matrix was constructed for the AHP hierarchy method study using the sum product method. The resulting weight values for the indicators C1–C23 are as follows: 5.021%, 5.230%, 5.439%, 2.510%, 4.549%, 5.423%, 3.520%, 2.929%, 4.812%, 5.021%, 5.858%, 3.138%, 4.812%, 3.138%, 3.679%, 4.393%, 5.648%, 3.975%, 4.554%, 4.393%, 5.23%, 4.602%, and 2.128%. The maximum eigenvalue of 23.000 was obtained by combining the eigenvectors, and the corresponding CI value was 0.000, indicating that the AHP method was valid and reliable for the evaluation of the child-friendliness of the community pocket parks.

**Table 5.** AHP indicator weightings based on indicator descriptions.

| Item | Eigenvectors | Weighting Values | Maximum Eigenvalue | CI Value |
|---|---|---|---|---|
| C1: Accessibility of facilities | 1.155 | 0.05021 | | |
| C2: Openness in the park | 1.203 | 0.05230 | | |
| C3: Walkability in the park | 1.251 | 0.05439 | | |
| C4: Environmental quality | 0.577 | 0.02510 | | |
| C5: Park surface green space area | 1.046 | 0.04549 | | |
| C6: Green space per capita | 1.247 | 0.05423 | | |
| C7: Number of parks | 0.810 | 0.03520 | | |
| C8: Diversity of play | 0.674 | 0.02929 | | |
| C9: Challenging to explore | 1.107 | 0.04812 | | |
| C10: Significance of learning | 1.155 | 0.05021 | | |
| C11: Health and safety | 1.347 | 0.05858 | | |
| C12: Freedom of movement | 0.722 | 0.03138 | 23.000 | 0.000 |
| C13: Nature affinity | 1.107 | 0.04812 | | |
| C14: Length of time spent in nature | 0.722 | 0.03138 | | |
| C15: Urbanization rate | 0.846 | 0.03679 | | |
| C16: Design aesthetics | 1.010 | 0.04393 | | |
| C17: Play facility planning | 1.299 | 0.05648 | | |
| C18: Diversity of natural attributes | 0.914 | 0.03975 | | |
| C19: Air quality (proportion of days with air quality at and better than Class 2) | 1.047 | 0.04554 | | |
| C20: Shareability | 1.010 | 0.04393 | | |
| C21: Public recreational space | 1.203 | 0.05230 | | |
| C22: Participatory | 1.059 | 0.04602 | | |
| C23: Age structure of children (% aged 0–14) | 0.489 | 0.02128 | | |

The AHP hierarchical analysis method used for weight calculation requires a consistency test analysis, which involves calculating both the Cl and RI values using Equations (2) and (3). The Cl value was computed, and the RI value can be obtained by consulting Table 6. For the 23-order judgment matrix constructed in this study, the random consistency RI

value of 1.646 can be obtained by referring to Table 6. All of the consistency evaluations of the judgment matrix satisfy the requirement of a CR value less than 0.1. This indicates that the judgment matrix in this study has passed the consistency test, and that the calculated weights are reliable and consistent. This confirms the validity and reliability of the AHP method used for the evaluation of the child-friendliness of the community pocket parks.

**Table 6.** Summary of the results of the consistency test.

| Maximum Characteristic Root | CI Value | RI Value | CR Value | Consistency Test Results |
|---|---|---|---|---|
| 23.000 | 0.000 | 1.646 | 0.000 | passed |

### 3.2.2. The Method of Entropy Weighting Is Employed to Determine the Weights

The objective weights for the evaluation indicators were established utilizing the entropy weighting method, which involved applying Equations (5)–(7). Table 7 presents the corresponding weight values for the entropy weighting calculations of C1–C23 in Table 1. Specifically, the weight values for each indicator are as follows: 3.44%, 3.31%, 3.05%, 7.54%, 3.28%, 2.62%, 6.47%, 3.13%, 3.54%, 3.44%, 3.31%, 3.05%, 7.54%, 3.28%, 2.62%, 6.47%, 3.13%, 3.54%, 3.44%, 2.74%, 7.91%, 3.29%, 7.33%, 5.41%, 3.83%, 1.83%, 4.66%, 3.53%, 3.83%, 3.31%, 4.60%, and 7.93%.

**Table 7.** Entropy-weighted TOPSIS indicator description weight values.

| Item | Information Entropy Value $e$ | Information Utility Value $d$ | Weighting $w_i$ |
|---|---|---|---|
| C1: Accessibility of facilities | 0.9149 | 0.0851 | 0.0344 |
| C2: Openness in the park | 0.9180 | 0.0820 | 0.0331 |
| C3: Walkability in the park | 0.9245 | 0.0755 | 0.0305 |
| C4: Environmental quality | 0.8135 | 0.1865 | 0.0754 |
| C5: Park surface green space area | 0.9189 | 0.0811 | 0.0328 |
| C6: Green space per capita | 0.9353 | 0.0647 | 0.0262 |
| C7: Number of parks | 0.8400 | 0.1600 | 0.0647 |
| C8: Diversity of play | 0.9226 | 0.0774 | 0.0313 |
| C9: Challenging to explore | 0.9123 | 0.0877 | 0.0354 |
| C10: Significance of learning | 0.9149 | 0.0851 | 0.0344 |
| C11: Health and safety | 0.9323 | 0.0677 | 0.0274 |
| C12: Freedom of movement | 0.8042 | 0.1958 | 0.0791 |
| C13: Nature affinity | 0.9185 | 0.0815 | 0.0329 |
| C14: Length of time spent in nature | 0.8187 | 0.1813 | 0.0733 |
| C15: Urbanization rate | 0.8663 | 0.1337 | 0.0541 |
| C16: Design aesthetics | 0.9054 | 0.0946 | 0.0383 |
| C17: Play facility planning | 0.9547 | 0.0453 | 0.0183 |
| C18: Diversity of natural attributes | 0.8848 | 0.1152 | 0.0466 |
| C19: Air quality (proportion of days with air quality at and better than Class 2) | 0.9127 | 0.0873 | 0.0353 |
| C20: Shareability | 0.9054 | 0.0946 | 0.0383 |
| C21: Public recreational space | 0.9180 | 0.0820 | 0.0331 |
| C22: Participatory | 0.8862 | 0.1138 | 0.0460 |
| C23: Age structure of children (% aged 0–14) | 0.8039 | 0.1961 | 0.0793 |

### 3.2.3. Entropy Weighting Method for Determining Weights

In the AHP-Entropy TOPSIS method, Equation (8) is utilized to determine the combined weight values for each program layer element in the objectives. This involves merging the AHP weights and entropy weights to achieve a more comprehensive and precise assessment of the relative significance of each objective. These weight values are presented in Table 8, indicating that the method is well-suited for its purpose. The results in the table show that the corresponding integrated weight values for C1–C23 are: 4.3119%, 4.3217%, 4.1413%, 4.7246%, 3.7249%, 3.547%, 5.6855%, 2.2887%, 4.2526%, 4.3119%, 4.007%,

6.1966% 3.9522%, 5.7422%, 4.9688%, 4.2003%, 2.5803%, 4.6243%, 4.0132%, 4.2003%, 4.3217%, 5.2848%, and 4.2128%.

**Table 8.** AHP-entropy-weighted TOPSIS indicator composite weight value.

| Item | $W_{AHP}$ | $W_{Entropy}$ | Combined Weighting Values $W_i$ |
|---|---|---|---|
| C1: Accessibility of facilities | 0.05021 | 0.0344 | 0.043119 |
| C2: Openness in the park | 0.0523 | 0.0331 | 0.043217 |
| C3: Walkability in the park | 0.05439 | 0.0305 | 0.041413 |
| C4: Environmental quality | 0.0251 | 0.0754 | 0.047246 |
| C5: Park surface green space area | 0.04549 | 0.0328 | 0.037249 |
| C6: Green space per capita | 0.05423 | 0.0262 | 0.03547 |
| C7: Number of parks | 0.0352 | 0.0647 | 0.056855 |
| C8: Diversity of play | 0.02929 | 0.0313 | 0.022887 |
| C9: Challenging to explore | 0.04812 | 0.0354 | 0.042526 |
| C10: Significance of learning | 0.05021 | 0.0344 | 0.043119 |
| C11: Health and safety | 0.05858 | 0.0274 | 0.04007 |
| C12: Freedom of movement | 0.03138 | 0.0791 | 0.061966 |
| C13: Nature affinity | 0.04812 | 0.0329 | 0.039522 |
| C14: Length of time spent in nature | 0.03138 | 0.0733 | 0.057422 |
| C15: Urbanization rate | 0.03679 | 0.0541 | 0.049688 |
| C16: Design aesthetics | 0.04393 | 0.0383 | 0.042003 |
| C17: Play facility planning | 0.05648 | 0.0183 | 0.025803 |
| C18: Diversity of natural attributes | 0.03975 | 0.0466 | 0.046243 |
| C19: Air quality (proportion of days with air quality at and better than Class 2) | 0.04554 | 0.0353 | 0.040132 |
| C20: Shareability | 0.04393 | 0.0383 | 0.042003 |
| C21: Public recreational space | 0.0523 | 0.0331 | 0.043217 |
| C22: Participatory | 0.04602 | 0.046 | 0.052848 |
| C23: Age structure of children (% aged 0–14) | 0.02128 | 0.0793 | 0.042128 |

### 3.3. TOPSIS Evaluation Model

To achieve a more intuitive, realistic, and comprehensive evaluation outcome for the model, two weighting techniques were employed. The evaluation indicators were assessed on a 1 to 5 scale, with higher scores indicating better implementation and lower scores indicating a need for improvement. The judgment matrix for the 10 sets of data was derived using Equation (9), and the proximity of the scored data to the ideal solution was computed using Equations (10)–(14), as indicated in Table 9.

**Table 9.** TOPSIS evaluation calculation results.

| Item | D+ | D− | C | Sort |
|---|---|---|---|---|
| 1 | 0.094 | 0.162 | 0.633 | 1 |
| 2 | 0.111 | 0.145 | 0.565 | 2 |
| 3 | 0.115 | 0.123 | 0.517 | 3 |
| 4 | 0.155 | 0.123 | 0.443 | 4 |
| 5 | 0.163 | 0.123 | 0.430 | 5 |
| 6 | 0.146 | 0.100 | 0.408 | 6 |
| 7 | 0.149 | 0.092 | 0.382 | 7 |
| 8 | 0.155 | 0.104 | 0.401 | 8 |
| 9 | 0.165 | 0.125 | 0.431 | 9 |
| 10 | 0.156 | 0.125 | 0.445 | 10 |

The C value, which indicates the proximity to the positive ideal solution, was utilized to assess the 10 sets of data. The highest C value of 0.633 was observed in the first set of data, indicating that it was the most reasonable among the 10 sets. By evaluating the scores of this set of data, the most appropriate evaluation result for the child-friendliness of the community pocket park can be obtained. This method ensured a more comprehensive and

accurate evaluation of the child-friendliness of the community pocket park and provided a solid basis for decision-making.

## 4. Discussion

This study has made a significant contribution to evaluating the child-friendliness of community pocket parks by using a combination of AHP and entropy-weighted TOPSIS. These techniques helped determine both the subjective and objective weights of relevant evaluation indicators, resulting in a more comprehensive and accurate evaluation of community pocket parks. The findings presented in Figure 5 demonstrate that all factors' combined weight values have a positive and substantial influence on residents' perceptions of commercial street quality improvements' effectiveness.

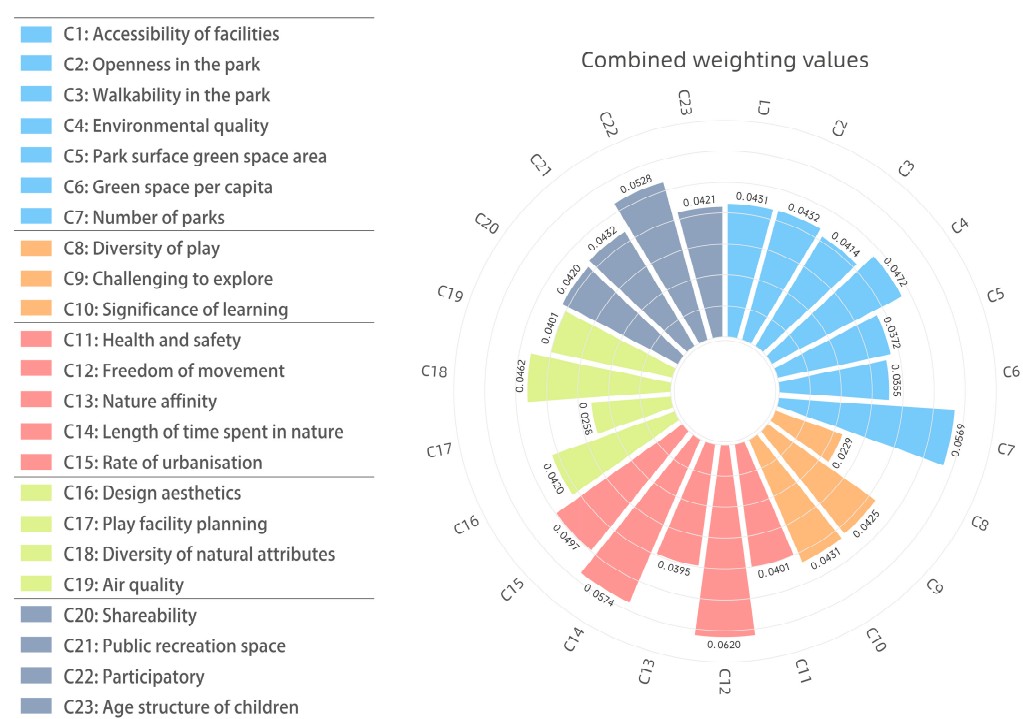

**Figure 5.** Histogram of combined weight values.

The indicators' relative significance is determined by their corresponding combined weight value, with freedom of movement (C12) having the greatest influence, followed by the number of parks (C7), participation (C22), and length of time spent in nature (C14). The remaining indicators have varying degrees of influence, with diversity of play (C8) and play facility planning (C17) having the lowest weighting and degree of influence. The next step is to analyze the results of the weighting process based on the relative importance of each evaluation indicator.

(1)  Freedom of movement (C12): In Chinese cities, parks are one of the few green public spaces available for children to engage in outdoor activities, which is vital for their healthy growth [49]. Pocket parks, being in the city's heart, provide a convenient and safe place for children to play and exercise freely in a familiar environment. Children's imagination is much richer than we think, and giving them more freedom of movement allows them to experience community pocket parks better, without necessarily having to be overly designed or expensive. Sand pits in playgrounds, for example, are full of magic to children, and piles of sand can keep them playing for half a day or long It has been found that sand areas are difficult-to-maintain facilities for washing bodies and that sand may increase the risk of inhalation of microplastics in the air [2,24]; however, the sand area was found to be a very popular play area for children in the survey.

(2) Number of parks (C7): In China, unclear standards, blind replication, and focus on either landscape or function have resulted in few parks that are genuinely attractive and appealing [67]. Thus, it is not solely a matter of quantity, but more of park quality. Land utilization should be improved, resources should be integrated, and pocket parks should be managed flexibly to promote actual utilization by children.

(3) Participatory (C22): Recreational activities with other children of different ages in community pocket parks and where good social interaction processes occur are key components of reflective participation, with a degree of social versatility possible for different age groups and contributing to the duration and intensity of physical activity [46]. Because children, teenagers, and older people like to have other people around, it helps to enliven the area and increase appreciation, safety and inclusiveness [57]. Participatory recreational activities with other children of different ages and good social processes are the main elements that reflect the participatory nature of pocket parks [47]. The focus should be on site planning and hands-on activities, including botanical landscaping, thematic exhibitions, designing a public space that allows different age groups to work together, and enabling children to experience the vitality of nature to achieve educational purposes.

(4) Hours spent in nature (C14): The number of hours children spend in the natural environment of a park depends on their interest in the natural world [51]. While the appeal of pocket parks to children is usually in the rides, the city is now more of a world of artificial objects, such as hard surfaces like tarmac, concrete, and houses made of glass and steel. In contrast, parks have more of a natural flavor, as they have relatively scarce plants and creatures. Therefore, for children, spending some time in a pocket park can help them gain a better exposure to and understanding of the natural world, increase their interest in and understanding of nature and, to some extent, can have an impact on children's social anxiety [50].

(5) C8: Play diversity and C17: Play facility planning were given a relatively low weighting and a low level of influence, with similar results to those of a study using Hong Kong as a high-density city [68]. There are specific children's play areas in community pocket parks that consist of vegetation and facilities that can significantly influence the intensity of children's activities [69]. In China there are specific factories that produce such facilities in large quantities, such as swings, balance beams, buckboards and play sets, resulting in a high degree of homogeneity in the quality of the facilities, where Western countries have higher standards. In fact, there is a serious tendency to simplify and conceptualize our perception of children, and in addition to traditional play facilities, wild play is also an important part of pocket parks [41]. Wild play can provide more diverse play to meet the needs of different children, while also promoting their physical and cognitive development. In wild play, children can improve their natural observation and cognitive skills through exploration and discovery of the natural environment, as well as their social skills and emotional development through interaction and cooperation.

## 5. Conclusions

Although remarkable progress has been made in constructing community pocket parks in China, there are still environmental problems to be addressed, such as serious homogenization and a lack of appeal to children in Hefei. To address these issues, the following recommendations are proposed:

(1) Build pocket parks that are tailored to local conditions. Design pocket parks based on the unique characteristics and needs of different areas to prevent homogenization. Ensure that basic facilities, such as leisure seats, children's play facilities, and open spaces, are appropriately equipped to meet the needs of the general public. Involve children in the construction and maintenance of pocket parks to obtain their feedback and improve the sustainability and long-term use value of the parks.

(2)  Design pocket parks according to standard systems related to their construction. Prioritize areas that are not covered by large parks during planning and site selection. Protect the original topography and landscape, large and old trees, and select native and suitable plants. Conduct special studies and surveys during the design stage to fully consider the needs and characteristics of children. Communicate and interact with children and parents to obtain their ideas and needs as a basis for design. Add parent-child play areas for younger children and challenging play facilities, such as climbing walls and trampolines, for older children.

(3)  Focus on planning and designing for wild play. Pay attention to the diversity and innovation of play facilities and design wildlife games. Incorporate natural elements, such as trees, rocks, and ponds, to provide more space for children to explore and create. Design wild play areas, such as nature exploration areas and teamwork areas, to allow children to have more fun and grow through free play and interaction.

This study used AHP-Entropy TOPSIS to evaluate the child-friendliness of community pocket parks in Hefei and provided planning and design suggestions to promote children's activities in pocket parks. This study can be extended to other cities and regions to compare the child-friendliness of pocket parks in different areas and provide a comprehensive reference for park planning and design. Future studies can explore ways to involve children in park planning and design and improve the sustainability of pocket parks by increasing green areas and improving self-maintenance.

**Author Contributions:** Conceptualization, L.Z. and X.X.; methodology, L.Z. and X.X.; software, X.X.; validation, L.Z., X.X. and Y.G.; formal analysis, Y.G.; investigation, L.Z. and X.X.; resources, X.X.; data curation, X.X.; writing—original draft preparation, X.X.; writing—review and editing, X.X. and Y.G.; visualization, L.Z. and X.X.; supervision, L.Z. and Y.G.; project administration, L.Z. and Y.G.; funding acquisition, L.Z. All authors have read and agreed to the published version of the manuscript.

**Funding:** The Anhui Provincial Philosophy and Social Sciences Planning Project supported this research (grant number AHSKQ2019D088).

**Institutional Review Board Statement:** The study was conducted in accordance with the Declaration of Helsinki, and approved by the Anhui Cultural Tourism Innovation Development Research Institute (protocol code ECACTIDRI-2023-001, and date of approval 2 February 2023).

**Informed Consent Statement:** All participants in the study provided their informed consent.

**Data Availability Statement:** If requested, the corresponding author can make the data presented in this study available.

**Conflicts of Interest:** The authors have no conflict of interest to declare.

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
