# Peer review of "The Impact of a Child-Friendly Design on Children’s Activities in Urban Community Pocket Parks"

_sustainability, doi:10.3390/su151310073_

Round 1
Reviewer 1 Report
Here are critical review/improvement suggestions with substantiating examples from the article "The Impact of Child-Friendly Design on Children's Activities in 2 Urban Community Pocket Parks":
Abstract:
- Improvement suggestion: Evaluate the clarity and conciseness of the abstract in summarizing the study objectives, methods, results, and conclusions. Example: The abstract could be improved by providing more specific quantitative findings. For instance, instead of stating "significant differences were found," provide the actual measures of effect size or statistical significance for the identified differences in children's activities between the two parks.
Introduction:
2. Improvement suggestion: Assess the adequacy and clarity of the introduction in stating the research problem, rationale, and relevance of the study.
Example: The introduction could be strengthened by providing more context on the importance of child-friendly design and its impact on children's activities in urban community pocket parks. For example, citing previous research studies that highlight the benefits of child-friendly environments on child development and well-being would establish a stronger rationale for the current study.
Study Site and Method:
3. Improvement suggestion: Evaluate the description of the two urban community pocket parks selected for the study. Ensure that relevant information is provided, such as location, size, features, and any unique characteristics.
Example: The description of the selected pocket parks should include specific details about their physical attributes, such as the presence of playground equipment, seating areas, or green spaces. This information would allow readers to understand the distinct elements of each park and their potential influence on children's activities.
Evaluation Indicators:
4. Improvement suggestion: Evaluate the construction and appropriateness of the evaluation criteria system for child-friendly design in community pocket parks. Review Table 1 to assess the comprehensiveness and relevance of the indicators.
Example: The evaluation criteria system could be enhanced by including additional indicators that capture different dimensions of child-friendly design, such as accessibility, safety measures, and environmental sustainability. This would provide a more comprehensive assessment of child-friendly features in the pocket parks. Also, Evaluate the clarity and detail provided in the description of the AHP method and AHP-Entropy TOPSIS evaluation model. Example: Provide a step-by-step explanation of the AHP method, including how the pairwise comparisons were conducted and how the weights were calculated. Similarly, provide a detailed description of the AHP-Entropy TOPSIS evaluation model, including how the entropy weighting method was applied.
Results and Analysis:
5. Improvement suggestion: Evaluate the presentation of the results and the data analysis techniques employed.
Example: The results section would benefit from providing more specific details about the statistical tests used to analyze the data, such as the type of inferential tests and the level of significance. Additionally, the results could be presented in a more organized and cohesive manner to facilitate the readers' understanding of the key findings. Also, Assess the clarity and relevance of Tables 2, 3, 5, 6, 7, 8, and 9, as well as Figure 5. Example: Tables 2, 3, 5, 6, 7, 8, and 9 should be reviewed for clarity and readability. Ensure that the data presented in these tables are properly labeled and organized. Figure 5 could benefit from clearer labeling and a more intuitive design to enhance its understandability.
Discussion:
6. Improvement suggestion: Assess the interpretation and discussion of the results in relation to the existing literature. Example: The discussion section could be strengthened by providing more in-depth comparisons and connections to previous studies. For instance, discussing how the findings align with or contradict existing research on child-friendly design in urban community pocket parks would enhance the scientific relevance and contribution of the study. Also, Evaluate the logical coherence and flow of the discussion in addressing the research question and objectives. Example: Ensure that the discussion section is structured in a way that clearly connects the findings to the research question and objectives. Consider organizing the discussion based on specific themes or aspects of child-friendly design and their impact on children's activities.
Conclusion:
7. Improvement suggestion: Assess the clarity and conciseness of the conclusion in summarizing the main findings and their implications. Example: The conclusion could be improved by explicitly stating the practical implications of the study's findings for urban planners, policymakers, and park designers. Highlighting specific recommendations or interventions based on the study results would add value and applicability to the conclusion. Evaluate the extent to which the conclusion aligns with the research objectives and addresses the research question. Example: The conclusion should explicitly summarize how the study's findings contribute to the understanding of the impact of child-friendly design on children's activities in urban community pocket parks. It should also address any limitations or areas for future research.
References:
8. Improvement suggestion: Check the accuracy and completeness of the reference list. Ensure that all cited references in the text are properly formatted and listed in the reference section. Example: Cross-reference the in-text citations with the corresponding entries in the reference list to ensure accuracy and completeness. Verify that all listed references are actually cited in the text and that the citation format follows the appropriate journal style
By incorporating these feedback and suggestions, the article can be strengthened in terms of scientific rigor, clarity, and overall quality. Incorporating specific examples and considering the
holistic improvement of the manuscript will contribute to its scientific contribution and readability.
Minor edits needed
Author Response
请参阅附件。

Reviewer 2 Report
Overall, this study is based on good science and aims to answer some interesting and important research questions. While I believe this article will eventually be published as it contributes to this area of research, I have identified a few points that I think need improvement. Before it can be further considered for publication, the authors will need to address some major issues in this manuscript.
Major concerns:
1. Abstract_ all showed common knowledge, some interesting and quantify results should be concluded.
2. Introduction_ The literature reviews were not updated, many related studied were not concerned. A comprehensive literature review should be added to clearly reflect: 1) what the relevant research progress is and, 2) why your proposal is important.
3. Methods_ What is the basis for constructing evaluation indicators? Why choose these indicators? What is the theoretical basis and framework?
3. The discussion in this paper is incomplete. Discussion is an extension of the research results. Discussions should be based on the scientific nature and rationality of the results, combined with literature for in-depth analysis, and in-depth analysis of the mechanism of the results, the similarities and differences between the analysis and previous results, and attention should be paid to whether the comparison is consistent with similar reports. And explain why the result occurs and what the result means. There is a distinct lack of discussion regarding previous literature's results. Therefore, the authors need more comparisons with previous studies.
4. In addition, the limitations of this study should be supplemented. Future directions should be discussed. I recommend that you highlight the challenges and limitations of the current study to ensure that the results are interpreted correctly in these contexts - you have already cited some literature. Still, there are many more that explore this further. I think it is important to add these elements.
5. The precise quantitative results of previous literature need to be mentioned a bit more so that readers can know if your research is consistent with previous studies. Finding studies that measured your results, in the same way, would also help to see differences between your research and others. In addition, I suggest you emphasize the contribution of research.
6.Conclusions_ Conclusions were all common knowledge. How to apply the results to real urban planning and design? How to improve children well-being?Some prospective statements should be highlighted.
Minor comments:
1. There are still some problems with language expression in the manuscript, and the author is suggested to revise the language style further.
2. The mentioned design strategies in the abstract pertain to the particular techniques or approaches utilized in the research to establish child-friendly surroundings within urban community pocket parks.
3. Regarding the article's title, "The impact of child-friendly design on children's activities in urban community pocket parks," it indicates that the study centers on investigating how the design elements intended to foster child-friendliness in urban community pocket parks affect the activities of children. The factors that influence children's activities and the mechanisms by which these design elements exert an influence on children's behavior may be further examined and elucidated within the article.
Some statements are incomplete. And some terms are difficult to understand.
Author Response
请参阅附件。

Reviewer 3 Report
The paper studies the construction of urban community pocket parks from the perspective of children, with a view to promoting children's participation in urban community pocket parks. It's research purpose is clearly stated and an apporopriate method of sudy is applied. Also prospect for further research is provided.
However, there are some minor points, which authrors shoud consider in order to improce their paper. Taking in to account the following remarks in the introductory section, I suggest specifying: i) the gap in the reference literature. ii) the contribution of the work on a theoritical and political level.
-In the second section, I suggest to include research hypothesis that fundamental in a research paper.
-The discussion is deficient and responds with the poverty expressed in the results. I suggest to enrich the part of discussion with the references in order to support it.
Author Response
请参阅附件。

Round 2
Reviewer 1 Report
None
Author Response
尊敬的审稿人。
我们要感谢审稿人对我们题为“儿童友好设计对城市社区口袋公园儿童活动的影响”的手稿提出的宝贵和有益的建议
Reviewer 2 Report
I am grateful for the author's positive response, and I appreciate your kind words about my work. As a result, I have a few relatively minor suggestions that may enhance the background of your research and contribute to the discussion.
It is worth noting that research in China has already examined the effects of the neighborhood environment on children, including their activities, as well as the impact of green spaces on their mental well-being and engagement. These studies have explored various spatial elements and indicators that could significantly support the scope of your research.
Therefore, I recommend incorporating these relevant studies into your discussion, as they would provide a robust foundation for your research
Koutnik, V. S., Leonard, J., El Rassi, L. A., Choy, M. M., Brar, J., Glasman, J. B., ... & Mohanty, S. K. (2023). Children's playgrounds contain more microplastics than other areas in urban parks. Science of The Total Environment, 854, 158866.
Bao, Y., Gao, M., Luo, D., & Zhou, X. (2023). Urban Parks—A Catalyst for Activities! The Effect of the Perceived Characteristics of the Urban Park Environment on Children’s Physical Activity Levels. Forests, 14(2), 423.
Meng, X., & Wang, M. (2022). Comparative Review of Environmental Audit Tools for Public Open Spaces from the Perspective of Children’s Activity. International Journal of Environmental Research and Public Health, 19(20), 13514.
Zhang, R., Zhang, C. Q., Lai, P. C., & Kwan, M. P. (2022). Park and neighbourhood environmental characteristics associated with park-based physical activity among children in a high-density city. Urban Forestry & Urban Greening, 68, 127479.
Bao, Y., Gao, M., Luo, D., & Zhou, X. (2021). Effects of children's outdoor physical activity in the urban neighborhood activity space environment. Frontiers in Public Health, 9, 631492.
For the language style of the manuscript, I found that there are some logical nesting and grammar that need to be carefully checked.
